# Flexible Symmetric-Defection Antenna with Bending and Thermal Insensitivity for Miniaturized UAV

**DOI:** 10.3390/mi15010159

**Published:** 2024-01-21

**Authors:** Xueli Nan, Tongtong Kang, Zhonghe Zhang, Xin Wang, Jiale Zhang, Yusheng Lei, Libo Gao, Jianli Cui, Hongcheng Xu

**Affiliations:** 1School of Automation and Software Engineering, Shanxi University, Taiyuan 030006, China; nanxueli@sxu.edu.cn (X.N.); 202123603013@email.sxu.edu.cn (T.K.); 202123601010@email.sxu.edu.cn (X.W.); 202123603033@email.sxu.edu.cn (J.Z.); 2School of Biomedical Engineering, Shanghai Jiao Tong University, Shanghai 200030, China; 3College of Electronics and Information Engineering, Shenzhen University, Shenzhen 518061, China; zzh422384683@163.com; 4Pen-Tung Sah Institute of Micro-Nano Science and Technology, Xiamen University, Xiamen 361102, China; 19920231151619@stu.xmu.edu.cn (Y.L.); lbgao@xmu.edu.cn (L.G.); 5School of Physics and Electronics Engineering, Yuncheng University, Yuncheng 044000, China; 6School of Instrument Science and Technology, Xi’an Jiaotong University, Xi’an 710049, China

**Keywords:** antenna, flexible, bend, thermal, insensitivity

## Abstract

Flexible conformal-enabled antennas have great potential for various developable surface-built unmanned aerial vehicles (UAVs) due to their superior mechanical compliance as well as maintaining excellent electromagnetic features. However, it remains a challenge that the antenna holds bending and thermal insensitivity to negligibly shift resonant frequency during conformal attachment and aerial flight, respectively. Here, we report a flexible symmetric-defection antenna (FSDA) with bending and thermal insensitivity. By engraving a symmetric defection on the reflective ground, the radiated unit attached to the soft polydimethylsiloxane (PDMS) makes the antenna resonate at the ISM microwave band (resonant frequency = 2.44 GHz) and conformal with a miniaturized UAV. The antenna is also insensitive to both the bending-conformal attachment (20 mm < r < 70 mm) and thermal radiation (20~100 °C) due to the symmetric peripheral-current field along the defection and the low-change thermal effect of the PDMS, respectively. Therefore, the antenna in a non-bending state almost keeps the same impedance matching and radiation when it is attached to a cylinder-back of a UAV. The flexible antenna with bending and thermal insensitivity will pave the way for more conformal or wrapping applications.

## 1. Introduction

Flexible antennas with superior mechanical compliance as well as maintaining excellent electromagnetic (EM) features are widely attached to developable or undevelopable surfaces for various applications [1,2,3,4,5], such as wirelessly conformal communication [6], motion monitoring [7,8], power transmission [9], etc. Especially, antennas with strain insensitivity are flexible or stretchable which makes them possibly adaptable to diverse complex deformations [10,11,12,13,14]. For instance, liquid metal can be poured into PDMS microchannels to prepare antennas that can withstand tensile variations [15,16,17], and the two-dimensional serpentine structure can endow the antenna with stretchability [18]. However, flexibility is a double-edged sword that always provides limited or inconstant properties of the antenna during bending attachment [19,20,21,22,23,24,25]. Currently, alternative methods for this problem such as structural design and improved manufacturing technology can be used to achieve robust EM features of flexible antennas during deformation. For example, a symmetric-defection structure is used to make the flexible antenna suitable for bending deformation, however, only for a large radius (100~200 mm) for multiple-input multiple-output (MIMO) applications [26]. Although the flexible antenna prepared by self-assembly technology effectively reduces the structure cracking or failure caused by deformation [27], the influence of bending and folding deformation on the EM feature for the antenna is also very obvious, and the fluctuation of performance is larger. Hence, there is an extreme need to tailor alternative antenna structures to offset bending impacts during deformation.

In addition, especially for long-term flight unmanned aerial vehicles (UAVs), ohm loss from inertial electrical components always causes some thermal interference, further weakening the antenna impedance matching [28]. Previous studies have verified the resonant frequency or working bandwidth shifts or mismatches with varying temperatures owing to the thermal sensitivity of the substrate [29,30,31,32,33,34]. For example, Tchafa et al. proposed a Rogers RT/duroid 5880-based copper antenna with 40.5 ppm/°C frequency changes in TM_001_ mode [35] and Xu et al. used a CNT-based gas-permeable and resilient bowtie antenna to achieve 0.54 MHz/°C thermal sensitivity [36]. Mitradip et al. proposed a flexible temperature sensor with a sensitivity of ~1.2%/°C by a printed chipless poly (3,4-ethylenedioxythiophene):polystyrene (PEDOT:PSS)-based antenna [37]. For wireless sensors, the frequency shift from thermal variation, thus providing temperature sensing, is crucial, but thermal interference is negative for remoting communication antennas. Reports about methods or technology to eliminate thermal effects for antennas are rare. Therefore, it is greatly urgent to endow antennas with low heat sensitivity.

Here, we present a flexible symmetric-defection antenna (FSDA) with bending and thermal insensitivity. By engraving a symmetric-defection structure, the antenna performance can remain stable while withstanding bending deformation, and the low-change thermal effect of the PDMS material makes the antenna insensitive to temperature changes and it can be seamlessly conformal with undevelopable surfaces. The antenna achieves impedance matching at 2.44 GHz (S_11_ = −22 dB) with a working bandwidth of 330 MHz and an omnidirectional radiation characteristic. Finally, the proposed FSDA maintains the same impedance matching and radiation while conformal with a miniaturized UAV, compared with the non-bending mode. This design provides a new idea for the study of conformal flexible antennas, which possibly represent potential applications for UAVs.

## 2. Design and Fabrication

### 2.1. Schematic and Design of the FSDA

The flexible antenna, which conforms with the vehicle surface of the UAV, is compact and has good aerodynamic properties [38,39,40], especially for operation between users and the base station at low altitude at the applied frequency band of ISM2.4 GHz [41,42,43,44], as shown in Figure 1a. The FSDA consists of the bottom symmetric-defection ground, the middle flexible dielectric substrate (PDMS, $\varepsilon_{r}$ = 2.65, tanδ = 0.031), and the top radiated unit (Figure 1b). Owing to the flexibility, the antenna can be seamlessly attached to the cylinder-back of a 3D-printed miniaturized UAV, as shown in Figure 1c. Figure 1d shows the process flow diagram of the antenna, in which a fast 355 nm UV laser with a power of 5 W at 50 kHz and a water-soluble tape (AQUASOL) are respectively used to engrave and transfer the copper-based structure. The laser-engraving and water-dissolving transferring method we adopted has vital advantages in the fabrication of various complex structures on the soft substrate and the speed (time consumption always < 1 min) for structure forming is superior to that of inkjet printing [45], screen printing [46], and 3D printing [47]. The equivalent circuit of the FSDA is shown in Figure 1e. $Z_{0}$ is the intrinsic impedance, and the LC parallel resonant circuit consists of a symmetric-defection structure loaded on the ground plane, where the circular slot is regarded as the inductance (L) and the middle gap is regarded as the capacitance (C). The dimension of the symmetric-defection structure can be optimized to match the input impedance. To be insensitive to bending deformation and thermal radiation, the antenna should hinder the equivalent electric length change during attachment and maintain the stable dielectric effect, as shown in Figure 1f, thereby providing slight frequency shifting which is crucial for practical applications of UAVs.

### 2.2. Parameter Optimization of the FSDA

Based on the slot antenna and microstrip antenna principles [48,49,50], the resonant frequency of antennas highly relates to the equivalent electrical length and the dielectric constant of substrate layer as follows:(1)$$L_{e}=\frac{v_{0}}{2f_{r}\sqrt{\varepsilon_{r}}}$$
where $L_{e}$ is the equivalent electrical length of the microstrip antenna, $v_{0}$ is the EM wave speed in vacuum, $\varepsilon_{r}$ is the dielectric constant of the substrate layer, $f_{r}$ is the resonant frequency of the antenna. The length of the patch is usually $\lambda_{g}/4$, where $\lambda_{g}$ is the working wavelength. We noted that the equivalent electrical length has a negative correlation with both $\varepsilon_{r}$ and $f_{r}$ as shown in Figure 2a, in terms of Equation (1).

In order to achieve robust aerodynamics for UAVs’ conformal antenna, a smaller electrical dimension effect provides low flight interference during long-term aircraft navigation at the ISM microwave band. Hence, the designed parameters in the inserted red rectangle are theoretically optimal for the simulated antenna. In addition, typically several soft materials [51,52], such as PDMS, TPU, and ecoflex silicone, have a dielectric constant of nearly 2~3. Further, according to the targeted regime, Figure 2b shows the designed structure dimension for the radiation unit and symmetric-defection ground. For a structural-defect microstrip antenna, the geometric dimension of the fed line and defection are crucial parameters to alter the impedance matching. To further optimize the antenna dimension, the return loss and VSWR are related to the stepped slot length (q), defection-circle radius ($R_{1}$), and their spacing ($W_{2}$), as shown in Figure 2c,d. Utilizing the finite element analysis (FEA) method for multi-parameter optimization, we note that both the $W_{2}$ and $R_{1}$ mainly determine the resonant frequency shift, and the q has an impact on the working bandwidth and resonant depth. By optimizing these parameters, we eventually obtain a definite $f_{r}$ and bandwidth. For an RF antenna device, when the return loss is less than −10 dB, the fed port can match the radiation unit at the corresponding working frequency, achieving the lowest power loss. In addition, four different antennas (Appendix A) were compared, and their return loss results are shown in Appendix A. We found that the proposed symmetric-defection antenna with a deeper S_11_ at the resonant frequency and a wider working bandwidth compared with others achieves superior impedance matching. The symmetric balun feeding line at the radiation unit and the symmetric defection at the ground make the antenna have deeper resonance and a wider working band. The final value of the geometric parameters of the FSDA is optimized, as shown in Table 1.

## 3. Results and Applications

### 3.1. EM Performance of the FSDA

The simulation and measurement results of the proposed antenna in free space are shown in Figure 3. The FSDA is fed and measured by the Siglent vector network analyzer (VNA, SIGLENT SVA 1032X) as shown in Figure 3a and the result of S_11_ is plotted in Figure 3b. It can be seen that the antenna achieves good matching at the resonant frequency of 2.44 GHz with an S_11_ value of −22 dB, and the working bandwidth is 330 MHz. The measured results are almost consistent with the simulation results ($f_{r}$ = 2.45 GHz, S_11_ = −23 dB, BW = 330 MHz). Figure 3c shows a good impedance match between the antenna and transmission line with an equivalent impedance of 50+j∗0, which helps to achieve highly efficient energy transmission. Figure 3d shows the near-field electric radiation, indicating a high-intensity electric field around the equivalent slotting edge in the FSDA. The 3D radiation field is shown in Figure 3e, and the peak gain at the resonance point is 1.6 dBi. The surface current distribution at 2.45 GHz is shown in Appendix A. It is visible that the current is mainly concentrated at the microstrip line, the main radiation gap of the radiating patch, and the symmetric-defection ground. And the currents on the radiating surface have similar flow paths around the edge of the patch, which is consistent with the theoretical prediction of the microstrip antenna edge radiation effect. Additionally, its radiation efficiency is 86% as shown in Appendix A, verifying its excellent radiation capability. As shown in Figure 3f, the H-plane radiation pattern is a circle, and the E-plane directional pattern is in the shape of an “∞”, showing good standard slot-like antenna radiation properties and a great agreement between the measurement and simulation results, further verifying the accuracy and reliability of the antenna.

### 3.2. Thermal Insensitivity

A UAV’s antenna should withstand the impact of thermal radiation owing to ohm loss from inertial electrical components. Thermal radiation results in the anisotropic thermal expansion of the flexible substrate material, thus increasing or decreasing the dielectric constant, further forming a definite frequency shift as follows [35]:(2)$$\frac{\delta f_{r}}{f_{r}}=-\frac{1}{2}\frac{\delta \varepsilon_{r}}{\varepsilon_{r}}-\frac{\delta L_{e}}{L_{e}},$$
(3)$$\frac{\delta \varepsilon_{r}}{\varepsilon_{r}}=\alpha_{\varepsilon_{r}}\delta T$$
where $\frac{\delta f_{r}}{f_{r}}$ is the normalized frequency shift, $\frac{\delta \varepsilon_{r}}{\varepsilon_{r}}$ is the normalized dielectric constant, $\frac{\delta L_{e}}{L_{e}}$ is the normalized equivalent length, $\alpha_{\varepsilon_{r}}$ is the thermal coefficient of dielectric constant (TCDk) of the substrate along the length direction. As for the simulated frequency response versus the dielectric constant and the equivalent dimension, within a special frequency shift range (<0.5 GHz) for typical soft material ($\varepsilon_{r}$ = 2~3), the low thermal sensitivity of the antenna should focus on the inserted red targeted regime as shown in Figure 4a.

When the FSDA is connected with the VNA (Figure 4b), it is heated by a thermal radiation machine, and the S_11_ parameter is measured for the antenna at different temperatures (Appendix A). Figure 4c shows the measurement results of the S_11_ in the temperature range from 20~100 °C, indicating that the antenna maintains a low reflection loss and exhibits good thermal stability. Figure 4d,e depict the variations of resonant frequency ($\Delta f_{r}$) and resonant depth with temperature, respectively. It can be seen that both the frequency shifting and resonance depth merely change at a negligible level (45 MHz and −4 dB), demonstrating its thermal stability. To elucidate the thermal-stability rationale of the FSDA, the thermal–magnetic co-simulation by the FEA method is built at a temperature range from 20 to 100 °C as shown in Figure 4f. When the thermal-field vector is directed from the input (fed end) to the output port, we note that the thermal radiation majorly occurs near the fed line, with only a little at the back of the circle defection (Figure 4g). This indicates that the thermal radiation can be attributed to ohm loss rather than EM radiation loss, and the volumetric heat-loss distribution of the FSDA further verified this property (Figure 4h). Figure 4i plots a quantitative relationship between the EM loss and the temperature variation. A value of less than 10% (0.0143 W variation) further indicates the low thermal–magnetic effect of the FSDA for thermal interference, verifying the insensitivity of the FSDA to temperature changes.

### 3.3. Bending Insensitivity

To be insensitive to bending deformation, the antenna should hinder the equivalent electric length change during attachment by this symmetric-defection structure. Similarly, in order to adapt the designed antenna to an irregularly curved surface, we simulated and measured the variation of return loss when the FSDA is attached to a cylinder with different radii (20~70 mm). Figure 5a shows the schematic diagram while the antenna conforms to the cylindrical surface. The simulated and measured results of S_11_ are shown in Figure 5b, where the results are almost consistent, with a good correlation in the target frequency band. The trends of resonant frequency and bandwidth with bending radius are shown in Figure 5c. The results reveal a small irregular shift in comparison to the flat state, as well as a slight increase in −10 dB bandwidth but the antenna remains stable in the desired frequency range. In addition, different degrees of bending deformation can affect the radiation performance of the antenna, as shown in Appendix A. The radiation efficiency under bending states is 80~85%, which is less than 5% different from the radiation efficiency under flat states. This indicates that the antenna can still maintain a high radiation ability in bending states.

We also conducted a uniaxial stretching test as shown in Appendix A. We observe that the working bandwidth and resonant frequency of the antenna show negligible changes for stretching up to 4% ($\Delta f_{r}$ < 10 MHz, ΔBW < 50 MHz, $\left|\Delta S_{11}\right|$ < 1 dB, Appendix A). This strain is attributed to the edge soft substrate instead of the copper-based antenna radiated unit, which maintains stable impedance matching. Furthermore, to verify its long-term durability, the antenna was stretched for 900 bending cycles (Figure 5d). It was observed that the antenna still returned to its original state, with no significant change in resonant frequency, further showing its excellent long-term durability. To further elaborate on the radiation performance of the bending insensitivity, Figure 5e,f illustrate the electric and magnetic field distributions in the flat and bending states, respectively. The field patterns are almost consistent, and the symmetric-deflection ground concentrates more current, verifying that the symmetric-deflection ground is able to provide bending insensitivity for the FSDA by balancing the currents during bending deformation in EM radiation.

In addition, we tested the return loss when the FSDA is bent in the E-plane as shown in Appendix A. We found that no observable frequency shifting occurs with the cylinder radius change, as shown in Appendix A. According to the E-field and H-field (Appendix A), the current distribution is almost consistent with that during the H-plane bending test, thus forming the negligible frequency shift. This is because the symmetric deflection on the ground concentrates more current, verifying that the symmetric-deflection ground provides bending insensitivity for the FSDA at the E-plane.

Also, the radiation pattern is carried out in the anechoic chamber far-field test system (KEYSIGHT E5071C, Dalian Dongshin Microwave, Dalian, China). Appendix A shows the simulated and measured radiation patterns when the antenna is in the flat and bending states (50 mm and 70 mm). The radiation patterns during bending testing show good agreement with the results from the flat antenna, which further verifies the insensitivity of the antenna to bending deformation. The E-plane radiation patterns have little deviations that may be attributed to the antenna’s installation error during the free space testing. These deviations are less than −20 dBi, verifying its inferior interference to the co-polarization radiation.

### 3.4. Aircraft Applications of the FSDA

We have further analyzed the performance of the FSDA on the UAV. Here, the FSDA is measured and simulated separately on the flat and cylinder-back surfaces of the UAV (Figure 6a). To simulate the real conductive UAV surface, we attached a copper foil with a thickness of 80 μm to the 3D-printed UAV surface. The measurement results are shown in Appendix A. The resonant frequency almost remains constant and the working bandwidth (<−10 dB) varies less than 10 MHz. These measurement results demonstrate that the proposed antenna can adapt to the UAV curved surface and has low sensitivity with more complex scenarios, which provides strong support for the further development of the flexible antenna in UAV applications. As can be seen in Figure 6b, when the antenna is fully attached to the cylinder-back of a 3D-printed miniaturized UAV (bending state), both the resonant frequency shift (${\Delta f}_{r}$) and working bandwidth change (ΔBW) at a negligible level (${\Delta f}_{r}$ = 70 MHz and ΔBW = 20 MHz). Overall, the measured result highly correlates with the simulated results, which are also consistent with the results described in Figure 5. Figure 6c depicts the realized gain curve. The maximum realized gain is 1.64 dBi at 2.44 GHz while the FSDA is conformal with the UAV, which shows that the antenna has good radiation performance. The simulated radiation direction patterns for the E-plane and H-plane in two states are shown in Figure 6d and the 3D radiation pattern of the FSDA is shown in Figure 6e. It is obvious that the bending has almost no effect on the radiation direction pattern and it still exhibits an omnidirectional radiation characteristic in the H-plane. Compared with previous literature, our antenna exhibits a lower relative bending frequency shift (2.8%, defined as $\frac{\left|F_{bending}-F_{flat}\right|}{F_{flat}}\%$), narrow return loss variation of only 1 dB, and 2% change in radiation efficiency as well as 4 dB change in thermal interference [27,37,53,54,55,56,57,58,59,60], as shown in Table 2. The above results fully demonstrate the reliability and effectiveness of the antenna in UAV communication systems.

## 4. Conclusions

In this work, a flexible symmetric-defection antenna was proposed for miniaturized UAV applications in the ISM microwave band. The antenna consists of a radiating unit, a symmetric-defection ground structure, and a PDMS substrate. An engraving–transferring method for the fabrication of a flexible antenna is low-cost and environmentally friendly compared with conventional PCB manufacturing process. The antenna achieves the resonant frequency at 2.44 GHz (S_11_ = −22 dB) with a working bandwidth of 330 MHz and omnidirectional radiation. The symmetric-defection ground balances current flowing during EM radiation as bending deformation and the PDMS-based substrate just provides a low-change thermal effect. Therefore, the proposed FSDA is highly flexible and insensitive to bending deformation and thermal radiation, making it suitable for applications in airborne devices such as UAVs. This antenna with bending and thermal insensitivity may be further optimized and integrated with various UAVs for high-fidelity communications.

## Figures and Tables

**Figure 1 micromachines-15-00159-f001:**
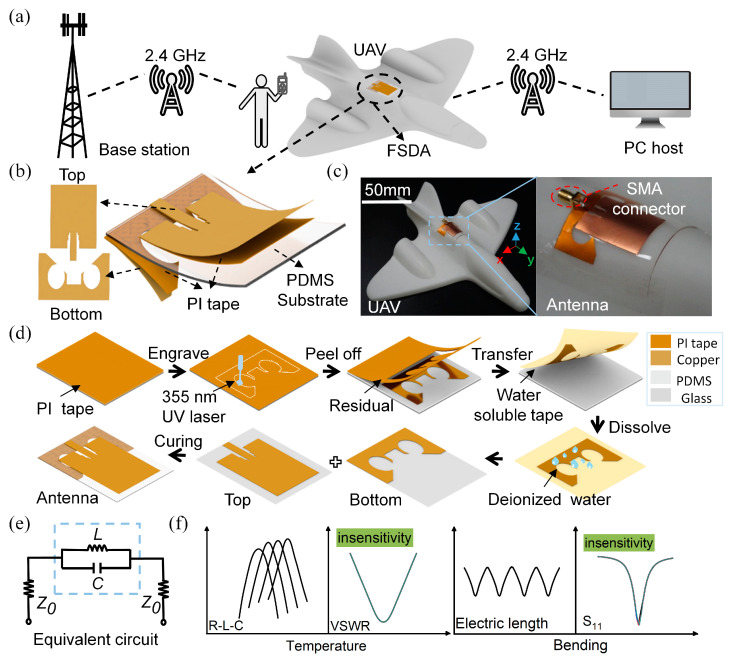
Rationale, design, and fabrication of the flexible symmetric-defection antenna (FSDA). (**a**) Wireless communication link between the FSDA attached to a UAV with radio frequency (RF) ends. (**b**) Three-dimensional structural illustration and its sectional view. (**c**) Photograph of the fabricated FSDA while it was conformal with a 3D-printed UAV model and the enlarged optical image. (**d**) Fabrication process of FSDA. (**e**) Equivalent RF resonant circuit. (**f**) Insensitive work rationale of the FSDA for thermal radiation and bending deformation.

**Figure 2 micromachines-15-00159-f002:**
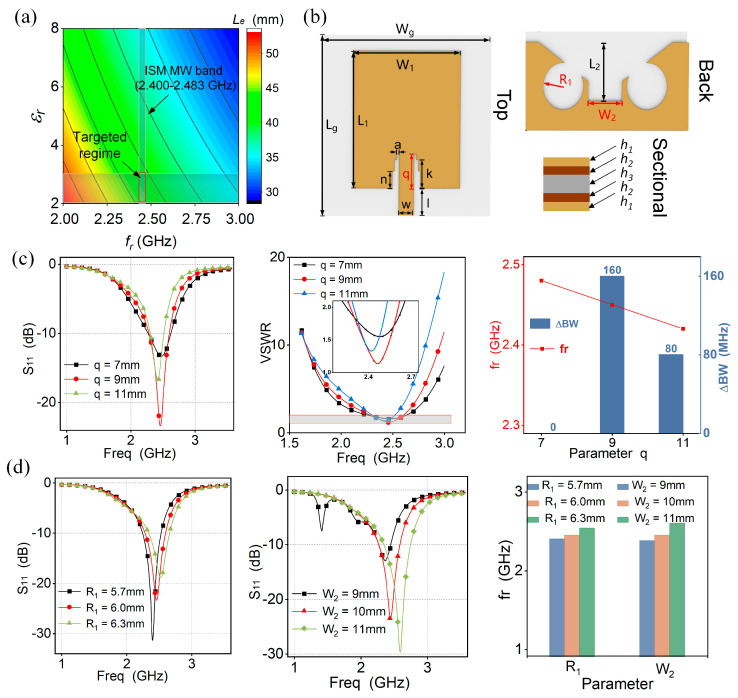
FSDA’s structure and parameter optimization. (**a**) Analytical relationship of equivalent length between dielectric constant and resonant frequency. (**b**) Three structural views. (**c**,**d**) Parameter optimization in the return loss to q, $R_{1}$, and $W_{2}$.

**Figure 3 micromachines-15-00159-f003:**
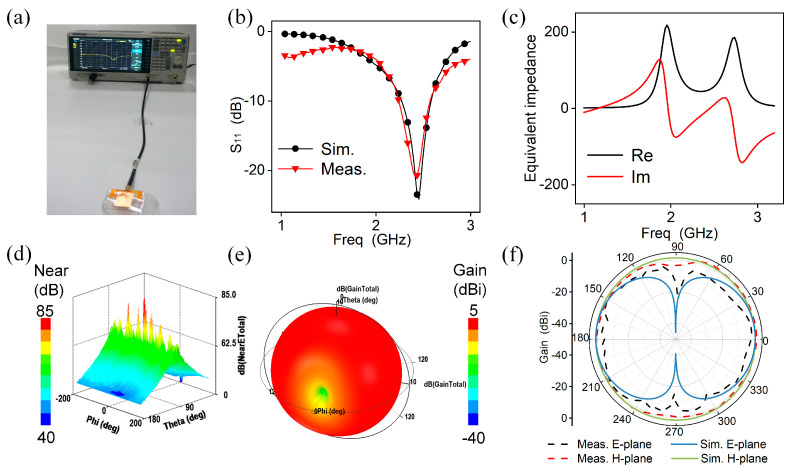
FSDA’s EM performance. (**a**) Test platform. (**b**) Simulated and measured S_11_ characteristics of the FSDA. (**c**) Equivalent impedance. (**d**) Near-field radiation. (**e**) Simulation result of 3D direction diagram. (**f**) Radiation patterns of H-plane and E-plane at resonant frequency when the antenna is in a flat state.

**Figure 4 micromachines-15-00159-f004:**
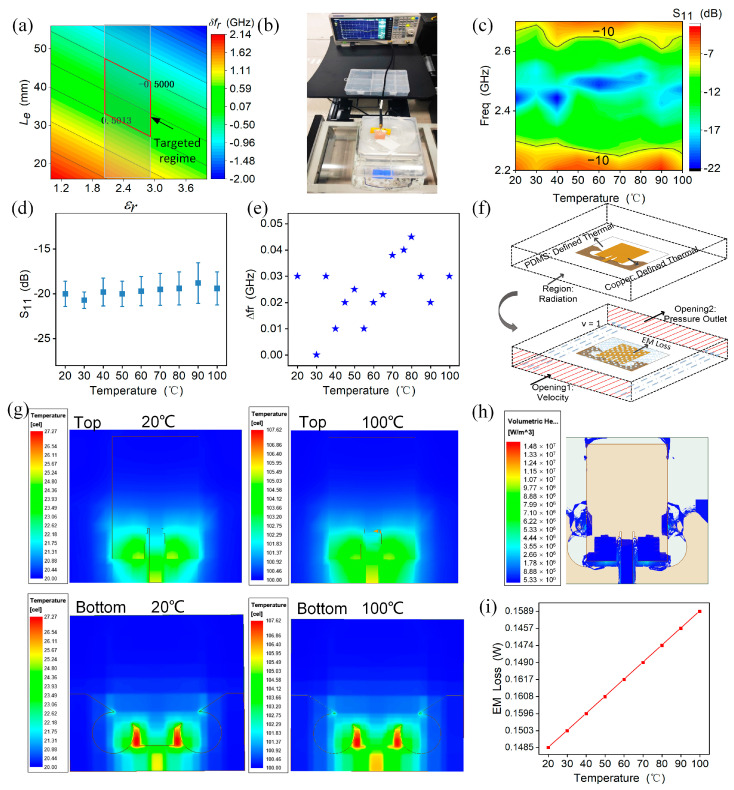
FSDA’s thermal-insensitivity verifications. (**a**) The relationship between normalized frequency shift and normalized dielectric constant and normalized equivalent length. (**b**) Temperature test platform. (**c**) Measured S_11_ result versus the frequency and temperature variations. (**d**,**e**) Variation of resonant frequency and resonant depth with temperature, respectively. (**f**) Schematic diagram of the thermal–magnetic co-simulation. (**g**) Thermal field of the FSDA under the thermal–magnetic coupling at 20 (left) and 100 °C (right). (**h**) Volumetric heat loss of the FSDA. (**i**) EM loss with varying temperatures.

**Figure 5 micromachines-15-00159-f005:**
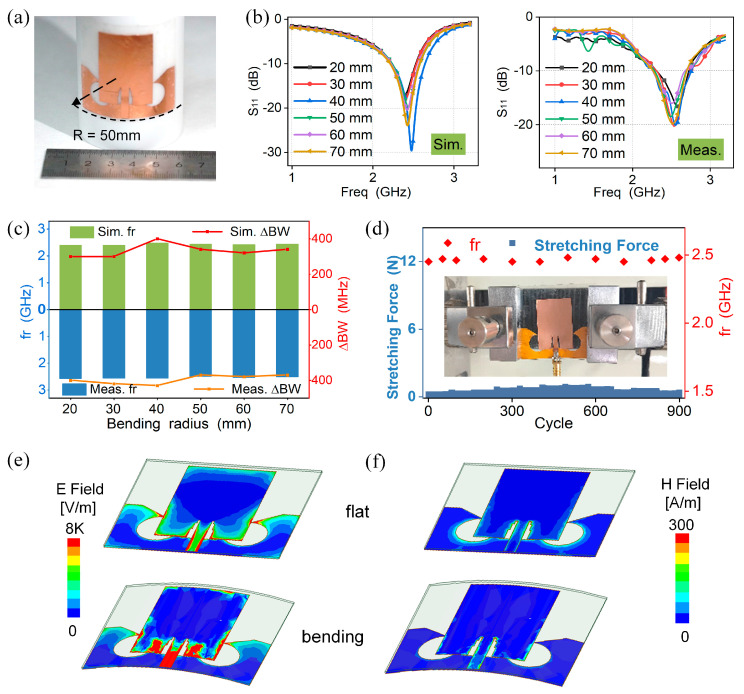
FSDA’s bending performance. (**a**) Bending conformity. (**b**) Simulated and measured results for S_11_ with different bending radii. (**c**) Variations of resonant frequency and bandwidth with bending radius. (**d**) Durability testing. (**e**,**f**) Electric and magnetic field comparison while the FSDA is flat (top) and bending (bottom).

**Figure 6 micromachines-15-00159-f006:**
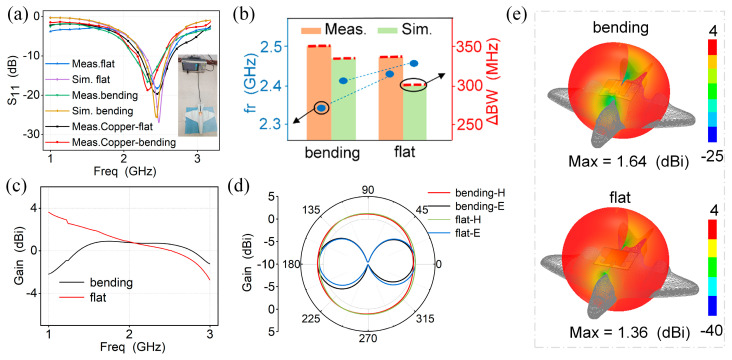
Applications of the FSDA with a no-load (flat) and bending state while attached to a UAV. (**a**) Comparison of simulated and measured S_11_ results. (**b**) Comparison of resonant frequency and bandwidth. (**c**) Gain curves with variation in frequency. (**d**) Radiation patterns of the H-plane and E-plane. (**e**) The 3D radiation pattern.

**Table 1 micromachines-15-00159-t001:** Values of main parameters of the FSDA.

Parameter	Value (mm)
$L_{g}$	45
$W_{g}$	49.9
$L_{1}$	36
$W_{1}$	25
l	7
w	3.5
$h_{1}$	0.008
$h_{2}$	0.05
$h_{3}$	0.54
a	0.51
n	4.5
k	7.5
q	9
$R_{1}$	6
$L_{2}$	15
$W_{2}$	10

**Table 2 micromachines-15-00159-t002:** Comparison of bending and thermal insensitivity of antennas.

Ref.	Flat				Bending						Thermal
	$f_{r}$(GHz)	S_11_ (dB)	∆BW(MHz)	Efficiency (%)	Radius/Angle	$\frac{\left|F_{bending}-F_{flat}\right|}{F_{flat}}$ (%)	$\left|∆S_{11}\right|$(dB)	|∆BW|(MHz)	Efficiency (%)	Durability	
[27]	2.32	−28	760	NA	40 (mm)	3%	10.84	310	NA	NA	NA
[37]	1.2/5.8	−13/−20	220/4000	NA	S_11_ > −10 dB at lateral bending, mismatching				NA	NA	~20% at 5.8 GHz
[53]	2.4	−21	275	25.8%	25 (mm)	4.4%	3	65	27.5%	NA	NA
[54]	2.45/5	−24/−14	170/700	37%/44%	28	2.1%	6/10	80/200	32%/36%	√	NA
[55]	2.53	−15	350	42.3%	70 (mm)	3.3%	2	190	NA	NA	NA
[56]	2.45	−22	128	30.5%	60°	0.4%	10	20	27.9%	NA	NA
[57]	5.7	−50	1600	80%	43	1.8%	24	100	83.5%	√	NA
[58]	2.43	−33	260	NA	40°	2%	19	80	NA	NA	NA
[59]	2.47	−21	250	83%	30 (mm)	3.6%	8	100	70%	NA	NA
[60]	2.43	−42	50	NA	43.2	0.8%	19	5	NA	√	NA
This work	2.45	−22	330	86%	40 (mm)	2.8%	1	100	84%	√	${\left|∆BW\right|}_{max}$~ 45 MHz, ${\left|∆S_{11}\right|}_{max}$~ 4 dB

√ yes; NA not applicable.

## Data Availability

The data that support the findings of this study are available on request from the author (X.N.) upon reasonable request.

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
