# Peer review of "Flexible Symmetric-Defection Antenna with Bending and Thermal Insensitivity for Miniaturized UAV"

_micromachines, 2024, doi:10.3390/mi15010159_

Round 1

Reviewer 1 Report

Comments and Suggestions for Authors

Flexible symmetric-defection antenna is proposed for UAV applications in the ISM band in this paper. The simulated results has the reason agreement between the measured results. The paper should be accepted for publication.

Reviewer 2 Report

Comments and Suggestions for Authors

The submitted manuscript on the Flexible Symmetric-Defection Antenna with Bending and 2 Thermal Insensitivity for Miniaturized UAV is promising and deserves attention.

However, to ensure the best possible reading experience for the readers, the figures in the manuscript should have a higher resolution.

Additionally, it would be constructive for the authors to highlight the efficiency of the presented antenna architecture and compare it with existing antennas that demonstrate similar performances.

This will help provide a more comprehensive analysis of the proposed solution and its potential impact in the field.

Comments on the Quality of English Language

To maintain scientific rigour, it is advisable to use concise sentences with precise language. This approach can help to ensure that your findings are presented clearly and accurately.

Reviewer 3 Report

Comments and Suggestions for Authors
  1. Comments to the authors:
  2. 1- I would appreciate further clarification on the novelty of your approach.

  3.  
  4. 2- At line 112, kindly cite the reference titled "Ultra wide band CPW-fed circularly polarized square slot antenna."

  5.  
  6. 3- In line 138, there appears to be an inconsistency in the figure numbering. Figure S1b is referenced but not present in the manuscript. Please verify and correct this discrepancy.

  7.  
  8. 4- Throughout the abstract and manuscript, emphasis is placed on the defection of ground impact on resonance frequency. However, considering conventional antennas, the primary determinant of resonance frequency is typically the radiator's (rectangular shape) length and the ground can only adjust the matching. Could you investigate and illustrate the influence of varying the rectangular patch length on the resonance frequency?

  9.  
  10. 5- It would be beneficial to include a presentation of the current distribution in your structure at the resonance frequency. This analysis would help identify the regions contributing significantly to the antenna's performance.

  11.  
  12. 6- Address the apparent contradiction in your antenna design. Monopole antennas are generally expected to provide UWB and omnidirectional radiation due to their partial ground. Please clarify how your antenna achieves a narrow bandwidth.

  13.  
  14. 7- Explain how did you select the length and width for the rectangular radiator.

  15.  
  16. 8- The resonance frequency of the antenna with a partial ground

  17. is influenced by the material of the structure it is mounted on. Considering this, why did you opt for a partial ground antenna instead of a full ground antenna, which typically offers a comparable bandwidth but with enhanced gain and directivity? Please provide clarification.

  18.  
  19. 9- Please include the model number of the VNA you used for S-parameters measurements. Also, do the same for the radiation pattern measurement system.

Comments on the Quality of English Language
  1. Comments to the authors:
  2. 1- I would appreciate further clarification on the novelty of your approach.

  3.  
  4. 2- At line 112, kindly cite the reference titled "Ultra wide band CPW-fed circularly polarized square slot antenna."

  5.  
  6. 3- In line 138, there appears to be an inconsistency in the figure numbering. Figure S1b is referenced but not present in the manuscript. Please verify and correct this discrepancy.

  7.  
  8. 4- Throughout the abstract and manuscript, emphasis is placed on the defection of ground impact on resonance frequency. However, considering conventional antennas, the primary determinant of resonance frequency is typically the radiator's (rectangular shape) length and the ground can only adjust the matching. Could you investigate and illustrate the influence of varying the rectangular patch length on the resonance frequency?

  9.  
  10. 5- It would be beneficial to include a presentation of the current distribution in your structure at the resonance frequency. This analysis would help identify the regions contributing significantly to the antenna's performance.

  11.  
  12. 6- Address the apparent contradiction in your antenna design. Monopole antennas are generally expected to provide UWB and omnidirectional radiation due to their partial ground. Please clarify how your antenna achieves a narrow bandwidth.

  13.  
  14. 7- Explain how did you select the length and width for the rectangular radiator.

  15.  
  16. 8- The resonance frequency of the antenna with a partial ground

  17. is influenced by the material of the structure it is mounted on. Considering this, why did you opt for a partial ground antenna instead of a full ground antenna, which typically offers a comparable bandwidth but with enhanced gain and directivity? Please provide clarification.

  18.  
  19. 9- Please include the model number of the VNA you used for S-parameters measurements. Also, do the same for the radiation pattern measurement system.

Round 2

Reviewer 3 Report

Comments and Suggestions for Authors

I don't have any additional remarks. I propose accepting the paper in its current state.